# Economic Analysis: Green Hydrogen Production Systems

**María Teresa Muñoz Díaz \*, Héctor Chávez Oróstica \*** 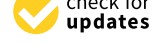 **and Javiera Guajardo**

Electrical Engineering Department, University of Santiago of Chile, Santiago 9170125, Chile; javiera.guajardo@usach.cl

\* Correspondence: maria.munoz.d@usach.cl (M.T.M.D.); hector.chavez@usach.cl (H.C.O.)

**Abstract:** The continued use of energy sources based on fossil fuels has various repercussions for the environment. These repercussions are being minimized through the use of renewable energy supplies and new techniques to decarbonize the global energy matrix. For many years, hydrogen has been one of the most used gases in all kinds of industry, and now it is possible to produce it efficiently, on a large scale, and in a non-polluting way. This gas is mainly used in the chemical industry and in the oil refining industry, but the constant growth of its applications has generated the interest of all the countries of the world. Its use in transportation, petrochemical industries, heating equipment, etc., will result in a decrease in the production of greenhouse gases, which are harmful to the environment. This means hydrogen is widely used and needed by countries, creating great opportunities for hydrogen export business. This paper details concepts about the production of green hydrogen, its associated technologies, and demand projections. In addition, the current situation of several countries regarding the use of this new fuel, their national strategy, and advances in research carried out in different parts of the world for various hydrogen generation projects are discussed. Additionally, the great opportunities that Chile has for this new hydrogen export business, thanks to the renewable energy production capacities in the north and south of the country, are discussed. The latter is key for countries that require large amounts of hydrogen to meet the demand from various industrial, energy, and transportation sectors. Therefore, it is of global importance to determine the real capacities that this country has in the face of this new green fuel. For this, modeling was carried out through mathematical representations, showing the behavior of the technologies involved in the production of hydrogen for a system composed of an on-grid photovoltaic plant, an electrolyser, and compressor, together with a storage system. The program optimized the capacities of the equipment in such a way as to reduce the costs of hydrogen production and thereby demonstrate Chile's capacity for the production of this fuel. From this, it was found that the LCOH for the case study was equivalent to 3.5 USD/kg, which is not yet considered a profitable value for the long term. Due to this, five case studies were analyzed, to see what factors influence the LCOH, and thereby reduce it as much as possible.

**Keywords:** hydrogen; renewable sources; hydrogen export; LCOH hydrogen; green hydrogen; hydrogen strategy



## 1. Introduction

The current mission of combating climate change, due to its devastating consequences, is present in most countries of the world. As such, a drastic reduction in the use of fossil fuels and greenhouse gas (GHG) emissions is expected. Much of generation of electricity, transportation, and heating is based on the burning of fossil fuels, producing one of the main polluting gases, carbon dioxide [1].

In 2019, the European Commission (EC) presented political initiatives to achieve zero GHG emissions by the year 2050 [2], mainly based on prioritizing the use of clean hydrogen for decarbonization. On the other hand, Chile joined this initiative, committing itself to achieving 70% of energy produced in the country from non-conventional renewable

energies (ERNC) [1]. The oil and gas industry is experiencing a historic crisis due to the environmental impact they have; however, green hydrogen intends to ameliorate the situation, as the fuel of the future [3].

### 1.1. Green Hydrogen

Hydrogen is the first chemical element on the periodic table, it is also the lightest and most abundant element in the universe [4]. It has the great advantage of being a carbon-less energy carrier, which can be produced by electrolysis of water or steam reforming of natural gas, coal, or biomass [1]. However, 96% is produced using fossil fuels [3], generating high GHG emissions.

It is possible to classify hydrogen by its production process, generating the well-known "hydrogen colors". Green hydrogen is generated from the electrolysis of water, where the electrolyser is powered by electricity produced by ERNC. On the other hand, black hydrogen bases its production on carbon, while gray hydrogen uses natural gas as raw material. There is also yellow hydrogen obtained from electricity of mixed origin or of nuclear origin. Finally, there is turquoise hydrogen, which uses natural gas or biomass [3].

Among all the methodologies used to generate hydrogen, the process with electrolysis of water is the best for producing this element quickly and with high purity [1]. However, this process demands a large amount of energy, representing its main disadvantage. In [1], it was shown that the production of hydrogen by means of methanol electrolysis allowed a saving of 60% of energy; in addition, they made a comparison between electrolysis processes with water, methanol, and hybrid sulfur.

There are multiple water electrolyser technologies, which differ mainly in their level of maturity, but only some are promising for use in the near future [5]. The alkaline water (ALK) electrolyser is one of the most basic and mature, and also has a 70% share of the green hydrogen market. On the other hand, the polymer electrolyte membrane (PEM) electrolysis shown in Figure 1 is being adopted by most major electrolyser manufacturers, as it produces higher quality hydrogen and can operate intermittently; however, it is expensive and has a lower production rate compared to ALK. Finally, the solid oxide electrolyser is still in the R&D stage, and although it has high efficiency at low cost, does not have a quick start-up [4].

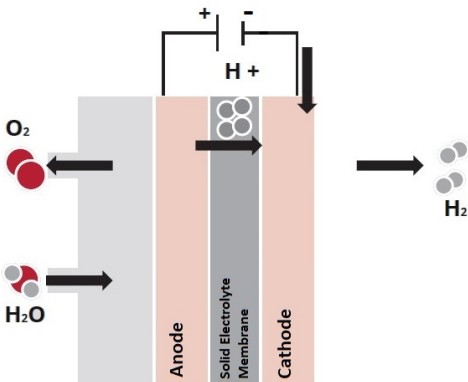

**Figure 1.** Polymeric electrolyte membrane technology [4].

### 1.2. Production Costs

D. Morales [6] mentions that one of the barriers to the production of green hydrogen is that its cost is higher than that of gray hydrogen and, furthermore, the cost of electricity is one of its major components. Currently, green hydrogen costs between 3.6 and 9.5 USD/kg in some regions of the world [7]. Gray hydrogen has the best profitability, followed by blue and finally green [8]. Consequently, large-scale production of green hydrogen is not yet profitable. However, it is expected that in the next few years the roles will reverse. In Figure 2, it can be seen that by the year 2050, green hydrogen will be more profitable than gray and blue hydrogen, both for production using ALK electrolysers and for PEM.

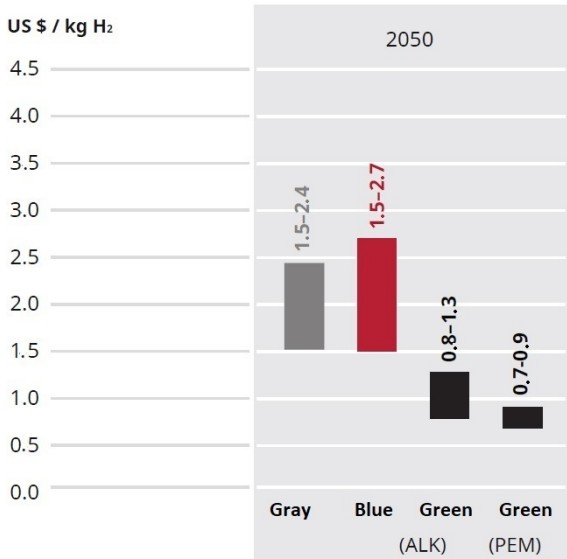

**Figure 2.** Evolution of the production cost of green hydrogen [4].

P. Ramírez [9] comments that the production costs of green hydrogen in Chile will always be significantly lower than in most countries in the world. The total installed power in the country is 25 GW, considering all energy sources, while the energy that could be obtained through solar energy is 2000 GW. Considering these figures, it is possible to cover domestic demand and still generate hydrogen exports.

Increasing production through the creation of electrolyser factories with large capacities will allow economies of scale to be generated, especially when the designs are standardized and the modules involved in production are optimized [5]. Considering the above, plus the decrease in energy generation costs, a possible significant reduction in production cost is presumed, making green hydrogen an economically viable solution [4].

The cost of transport is directly related to the distance, volume transported, and the mode of transport selected. If this represents a very high value, it will not be economically sustainable to transport green hydrogen. Under this scenario, it is considered convenient to locate the electrolysers near the points of demand. In Figure 3, graph is shown that relates distances, costs, and modes of transport. For short distances, it is advisable to use trucks to transport hydrogen in a liquid state compared to a gaseous one, since the liquid form has a higher energy density [10]. On the other hand, transportation through pipelines entails minimal costs and depends closely on distance and flow, where the latter allows generating economies of scale. However, reusing hydrogen pipelines requires a lot of capital [5].

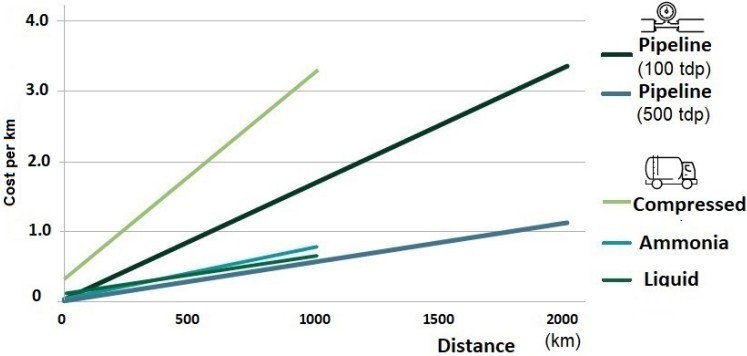

**Figure 3.** Transport costs by type and distance [11].

The solar committee in the report prepared by Tractebel [12], calculated an LCOH equal to 1.3–2.86 USD/kg in the year 2035. Regarding the IRENA reference [13], this describes the production of hydrogen in Chile from photovoltaics and concentrated solar power

(CSP) combined, which would lead to a leveled cost of hydrogen of around 2.7 USD/kg, based on IRENA assumptions. In addition, he explained that CORFO calculated an LCOH of 1.6 USD/kg for the year 2025 in its energy model connected to the grid (on-grid). The International Energy Agency (IEA) analyzed the production costs of hydrogen from different countries, including Chile, where it estimated an LCOH of 1.6–3 USD/kg from renewable energy [14]. Finally, in reference [15], the cost of the most competitive production system reached an LCOH of 1.67 USD/kg, with a cost reduction of around 25% by the year 2025. Despite the above, in 2020, the production of hydrogen from renewable energy was within the range of 3.0 to 7.5 USD/kg of the leveled cost of hydrogen [16].

## 2. Methods

Research has been undertaken that analyzed the feasibility of NCRE-based hydrogen production in different parts of the world. In [17], an infrastructure based on an offshore wind farms was studied, seeking to demonstrate that the industry is well established and distributed in the regions of France. The case study considered a plant along the Pays de la Loire coast and had an electrolyser, a 100-bar compressor, and a vessel in charge of transporting cylinders of compressed hydrogen, plus two sectors for storage of tanks located near the plant. The GAMS software was used to optimize the operation of the plant by minimizing the installed wind–hydrogen capacity. From this it was deduced that the optimal size of the park was inversely proportional to the storage capacity; therefore, a balance must be sought that avoids oversizing equipment such as the compressor and electrolyser, where the efficiency of the latter has been identified as a key issue in many studies regarding the profitability of hydrogen [18].

In [19], it was commented that a strategy to take advantage of the excess energy in photovoltaic systems would be to use batteries, which also allows maximizing the reliability of the system. However, the useful life of these devices is affected by the intermittency of photovoltaic energy; in addition, it presents a low efficiency and high self-discharge. The research proposed using a hybrid storage system based on batteries and hydrogen generated at times of overgeneration during the day. Based on a multi-objective algorithm, it was verified that the design produced benefits in terms of the environment, reliability, and economy, where aspects such as the location of the plant and the size of the equipment involved directly affected the profitability.

Based on the above, it can be seen that Chile is a potential large-scale green hydrogen exporter, thanks to the amount of solar resources that exist in the Atacama desert, as well as the guaranteed winds in the extreme south of the country [9]. Therefore, this research focused on analyzing the possible opportunities that the country has in terms of a hydrogen production plant that uses electrical energy from an on-grid photovoltaic plant. The capacity that Chile has regarding this new green fuel is of global importance, since it allows countries to obtain relevant information to generate import business or collaboration agreements with the country. With this information, it will be possible to generate a green hydrogen market that is sustainable over time, because by increasing the number of countries that have information on this fuel, it is possible that they will increase the decarbonization of their energy networks.

### 2.1. Optimization Model

In [1], the first approach to the optimization of the leveled cost of hydrogen (LCOH) was investigated through a mathematical model for a photovoltaic hydrogen generation system and for its subsequent export using the port of Mejillones, Antofagasta. The process is based on producing hydrogen from photovoltaic generation and power from the grid when required, such as on cloudy days or at night. Subsequently, there is the electrolyser module, which has as input the amount of water and electricity necessary for the production of hydrogen and for its subsequent compression and storage. Finally, the transfer is carried out for subsequent export. In Figure 4, representative scheme of each optimized module is shown. In [18], it was concluded that for cases of larger systems, lower photovoltaic capital

costs, and high prices of the electrical network, it is better to use a system that does not require connection to the electrical network.

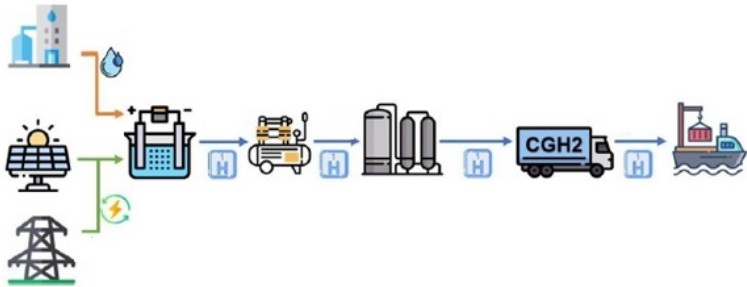

**Figure 4.** Representation of the modules of the optimized plant [1].

The objective function for minimizing presented in (1) considers the initial investment plus the total of the fixed and variable costs in the numerator, while in the denominator there is the total hydrogen production [1,18]. The economic value of the project considered an evaluation horizon of 20 years, together with a discount rate of 10 %. Table 1 presents the description of the variables of the equation.

$$min \frac{I_o + \sum_{t=1}^{n} \frac{C^{fijo} + C^{Var}}{(1+i)^t}}{\sum_{t=1}^{n} \frac{H}{(1+i)^t}} \tag{1}$$

The initial investment (2) considers the individual costs for the capacity of each module in the system. Among them is the photovoltaic, and the production of hydrogen by the electrolyser and compressor, plus the storage and transport systems. In addition, the variables are described in Table 2.

$$I_o = C^{PV} \cdot P^{PV} + C^{elect} \cdot P^{elect} + C^{CGH2} \cdot T^{CGH2} \tag{2}$$

On the other hand, with respect to operational costs, that is, those necessary for the operation of the project, such as labor costs, raw materials involved, and expenses directly related to production; it is possible to identify variable and fixed costs. Variable costs (3) are associated with the cost of energy from the electrical network (PPA), plus the water used by the electrolyser, based on a period "$k$" (hours), while fixed costs (4) correspond to operation and maintenance. A description of the variables presented below is shown in Table 3.

$$C_{Var} = \sum_{k} (E_k^{red} \cdot PPA) + \sum_{k} (C^{H2O} \cdot C^{sp,H2O} \cdot \dot{m}_k^{H2}) \tag{3}$$

$$C_{fijo} = O\&M \tag{4}$$

**Table 1.** Variables associated with the cost of hydrogen production.

| Variable | Description |
|---|---|
| $I_o$ | Initial investment (USD) |
| $C^{fixed}$ | Fixed operational costs (USD) |
| $C^{Var}$ | Variable operating costs (USD) |
| H | Hydrogen production per year (ton) |
| n | Total investment time (years) |
| i | Discount rate (%) |
| t | Period (years) |

**Table 2.** Variables associated with the initial investment.

| Variable | Description |
|---|---|
| $C^{PV}$ | Solar panel investment cost (USD/kW) |
| $P^{PV}$ | Capacity of the photovoltaic system (kW) |
| $C^{elect}$ | Electrolyser–compressor investment cost (USD/kW) |
| $P^{elect}$ | Capacity of the electrolyser–compressor system (kW) |
| $C^{CGH2}$ | Investment cost of hydrogen tank (USD/kg) |
| $T^{CGH2}$ | Hydrogen tank size (kg) |

**Table 3.** Variables associated with fixed and variable costs.

| Variable | Description |
|---|---|
| $E_k^{grid}$ | Energy from the grid delivered to the electrolyser in a period k (kWh) |
| $C^{H2O}$ | Specific cost of water (USD/L) |
| $C^{sp,H2O}$ | Specific water consumption (L/kg) |
| $\dot{m}_k^{H2}$ | Flow of hydrogen produced by electrolyser in period k (kg/h) |

This seeks to optimize production costs by determining the dimensions of each module based on their cost–benefit.

*2.2. Case Study*

This investigation considers a work horizon of an average meteorological year, obtaining results for 8760 h. In the first place, electricity is obtained from the electrical network, with a price equivalent to 63 (USD/MWh), together with that produced by the photovoltaic system, which considers radiation for a 1 MW photovoltaic plant, located in the Atacama desert, Antofagasta Region; with the characteristics specified in Table 4, obtained from the solar operator.

**Table 4.** Technical and economic characteristics of the photovoltaic plant.

| Characteristics | Value | Unit of Measure |
|---|---|---|
| Installed capacity | 1 | MW |
| Panel temperature coefficient | −0.45 | %/°C |
| Tilt | 23° | - |
| Inverter efficiency | 96 | % |
| Loss factor of the photovoltaic system | 14 | % |
| Plant factor | 35 | % |
| Annual generated energy | 3.04 | % |
| CAPEX photovoltaic plant | 740 | USD |
| OPEX photovoltaic plant | 1.7 | % CAPEX |

On the other hand, an alkaline type water electrolyser is considered, together with a compressor for the generation of gaseous hydrogen. Table 5 shows the characteristics associated with these technologies.

Additionally, a steel storage system is used for compressed hydrogen up to 350 bar, with a CAPEX and OPEX equivalent to 500 USD/kg and 2% CAPEX, respectively. The system is dimensioned with the objective of producing hydrogen capable of supplying a demand equivalent to 16 kton per year; that is, approximately 44 tons per day. Due to these levels of demand being considered low, it is recommended to transport hydrogen by truck. This means implies an efficiency of 98% associated with transportation losses.

**Table 5.** Technical and economic characteristics of the electrolyser–compressor.

| Characteristics | Value | Unit of Measure |
|---|---|---|
| Electrolyser specific energy consumption | 54 | kWh/kg |
| Compressor specific energy consumption | 1.36 | kWh/kg |
| Specific water consumption electrolyser | 8.9 | L/kg |
| CAPEX electrolyser–compressor | 600 | USD/kW |
| OPEX electrolyser–compressor | 2 | % of CAPEX |
| Cost of water | 0.003 | USD/L |

## 3. Results

The AIMMS program minimizes the objective function by analyzing the CAPEX, OPEX, and variable costs for each hour ("k"), delivering the capacity of the technologies involved. Once the aforementioned parameters have been replaced, Equation (5) is calculated, which represents the final version of the objective function used in the AIMMS program.

$$740 \cdot P^{pv} + 600 \cdot P^{elect} + 500 \cdot T^{CGH2} + \sum_{t=1}^{20} \frac{OPEX + 0.063 \cdot E_k^{red} + 0.003 \cdot 8.9 \cdot \dot{m}_k^{H2}}{(1+10\%)^t} \quad (5)$$

For the case study, the program shows that the photovoltaic plant must have a capacity of 180,436 MW, while the electrolyser and storage system are 171,314 MW and 11,327 tons, respectively. These values are shown in the "page manager" interface, designed in such a way that it shows the results numerically and graphically, as shown in Figure 5.

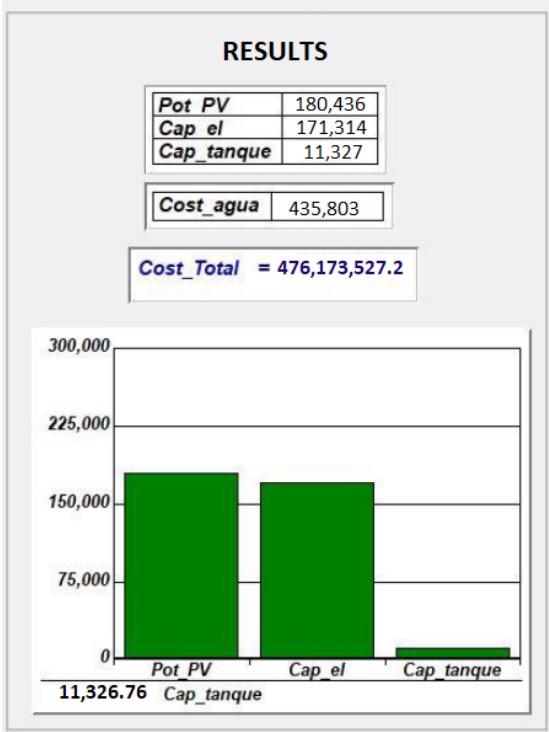

**Figure 5.** Results of the case study.

The costs associated with the system, considering the capabilities mentioned above, are broken down in Table 6.

**Table 6.** Costs in USD per concept for the case study.

| Concept | Formulation | Value |
|---|---|---|
| System PV | $C^{pv} \cdot P^{PV}$ | 133,522,425 |
| OPEX PV | $1.7\% \cdot C^{pv} \cdot P^{PV}$ | 2,269,881 |
| Electrical grid | $\sum \left( E_k^{grid} \cdot PPA \right)$ | 229,324,902 |
| Electrolyser system | $C^{elect} \cdot P^{elect}$ | 102,788,107 |
| OPEX electrolyser | $2\% \cdot C^{elect} \cdot P^{elect}$ | 2,055,762 |
| Water | $\sum \left( C^{H2O} \cdot C^{sp.H2O} \cdot \dot{m}_k^{H2} \right)$ | 435,803 |
| Storage system | $C^{CGH2} \cdot T^{CGH2}$ | 5,663,381 |
| OPEX storage | $2\% \cdot C^{CGH2} \cdot T^{CGH2}$ | 113,268 |
| Total cost of the system | - | 476,173,527 |

On the other hand, the net present value of hydrogen production was obtained, considering Equation (6), together with a hydrogen production *H* of 15,995,760, a discount rate *i* equivalent to 10%, and a period *n* of 20 years.

$$VAN_{H2} = \sum_{t=1}^{n} \frac{H}{(1+i)^t} \tag{6}$$

From this, it was found that the GO *H*2 is equivalent to 136,180,922 kg. Finally, the levelized cost of hydrogen (LCOH) was calculated for the case study, considering the total cost of the system together with the NPV of hydrogen production, as observed in the equation:

$$LCOH = \frac{I_o + \sum_{t=1}^{n} \frac{C^{Fijo} + C^{Variable}}{(1+i)^t}}{\sum_{t=1}^{n} \frac{H}{(1+i)^t}} \tag{7}$$

By substituting the values, it follows that the LCOH for the case study is equivalent to 3.5 USD/kg. Additionally, a sensitivity analysis of the model was carried out, where the system parameters were evaluated, to verify their impact.

### 3.1. Case Study 1: CAPEX PV Variation

In the first place, a decrease in the cost of the photovoltaic plant was analyzed. For this, it was considered that the CAPEX decreased by 16% with respect to the base case; that is, equivalent to 618 USD/kW. From this it was found that the photovoltaic plant requires a capacity of 246.61 MW, the electrolyser of 234.15 MW, and the storage capacity of 24.37 tons. This translates to a total cost of USD 422,811,303. As the amount of hydrogen produced is maintained, the NPV *H*2 is equivalent to 136,180,922 kg. With this, it was shown that the LCOH for this case study is 3.25 USD/kg.

In Figure 6, it can be seen that the capacity increased with respect to the base case, while the LCOH decreased. This is attributed to the fact that the consumption of electrical energy coming from the network decreased, since the implementation of the plant would imply a lower cost. Similarly, the electrolyser system increased its capacity by 36.7%, while the storage doubled.

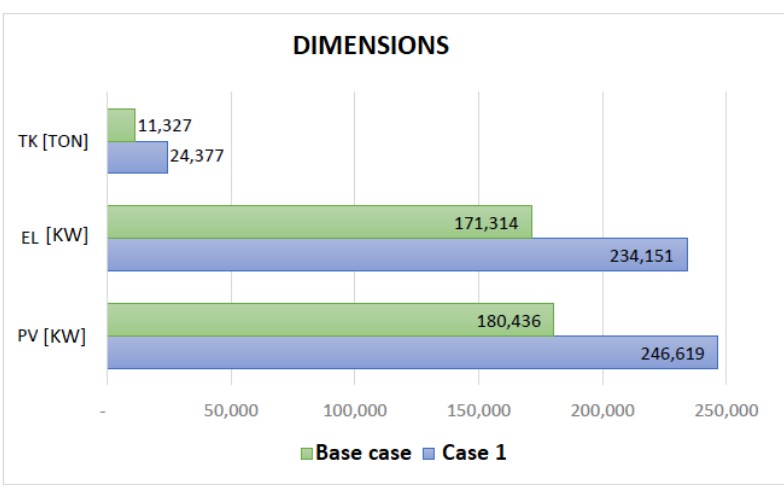

**Figure 6.** Comparison base case and case 1.

### 3.2. Case Study 2: PPP Variation

At the same time, the impact of a decrease in the cost of electricity from the electrical network was analyzed. For this, a cost equivalent to 54 USD/MWh was considered. The program showed that the capacities of the photovoltaic plant and the electrolyser are 108,643 kW and 103,150 kW, respectively. With this, it can be seen that both capacities decreased with respect to the base case, by approximately 37%. Additionally, the AIMMS showed that for this case study no storage system was required; therefore, the capacity was 0 kg. It was shown that the total cost of the system was 432,908,181.5 USD.

As the amount of hydrogen production was maintained, the NPV $H2$ was equivalent to 136,180,922 kg. As such, it was shown that the LCOH for this case study was 3.18 USD/kg.

In Figure 7, it can be seen that the capacity and LCOH decreased with respect to the base case. This is because the energy coming from the electrical network supplied all the energy in the hours when there was no radiation. However, the energy demand of the electrolyser remained constant, which is equivalent to the maximum power it consumed from the network. Additionally, it is noted that a storage system is not required.

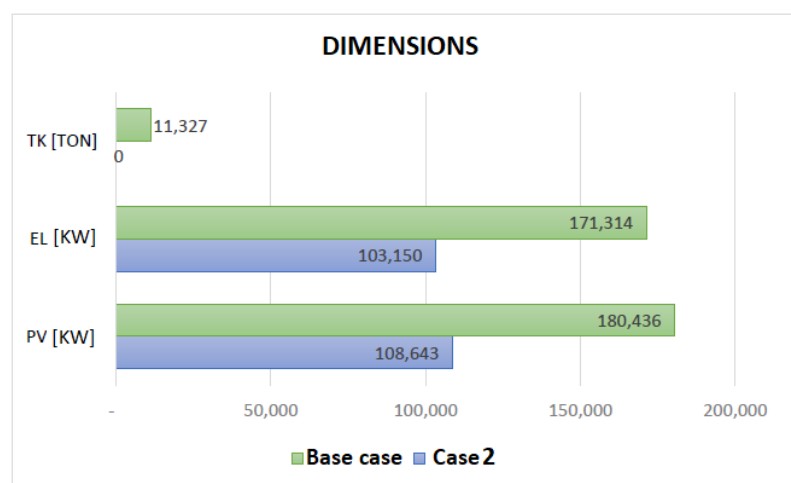

**Figure 7.** Comparison base case and case 2.

### 3.3. Case Study 3: Electrolyser CAPEX Variation

Additionally, the consequence of reducing the CAPEX of the electrolyser system to 340 USD/kW was studied. The program showed that the photovoltaic plant had a capacity of 247.33 MW, the electrolyser of 234.82 MW, and the storage system a capacity of 24,771 tons; obtaining a total cost of 405,961,863 USD. Considering that the amount of

hydrogen production was maintained, the NPV *H*2 was equivalent to 136,180,922 kg. With this, it was found that the LCOH for this case study was 2.98 USD/kg.

In Figure 8, it can be seen that the capacity increased with respect to the base case, while the LCOH decreased considerably. There was a notable increase in the solar plant, due to the fact that there was a growth in solar generation and a great decrease in the use of electricity from the network. In the same way, the electrolyser system increased its capacity by 37%, while the storage increased its capacity twice with respect to the base case.

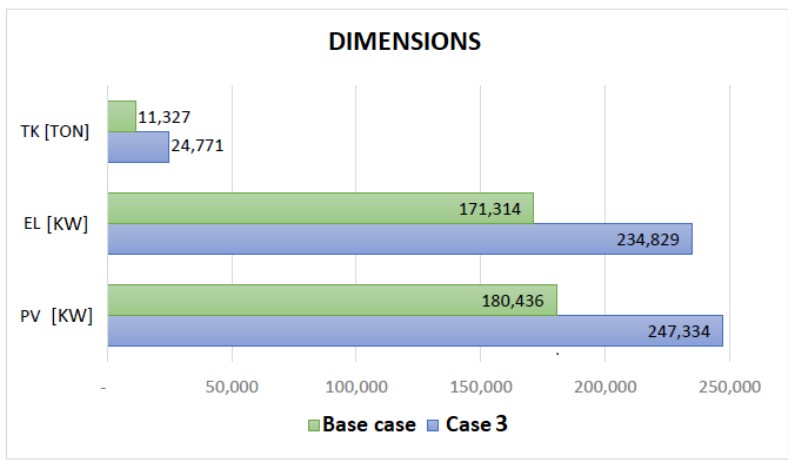

**Figure 8.** Base case and case 3 comparison.

### 3.4. Case Study 4: Storage CAPEX Variation

On the other hand, a reduction of the CAPEX by 20% for the storage system through gaseous compressed hydrogen tanks was studied. For this, a CAPEX of 400 USD/kg was considered. In this case study, it can be seen that the solar plant required a capacity of approximately 243 MW, the electrolyser of 230 MW, and the storage tanks of 23 tons. This led to a total system cost equivalent to USD 474,119,739. For this case, the hydrogen production was maintained; therefore, the NPV *H*2 was equivalent to 136,180,922 kg. As such, it was demonstrated that the LCOH for this case study was 3.48 USD/kg.

Although Figure 9 shows that the capacities of the photovoltaic system and the electrolyser increased by approximately 35% with respect to the base case, and the storage doubled, the LCOH did not decrease much. Therefore, the reduction of storage CAPEX did not greatly influence the LCOH.

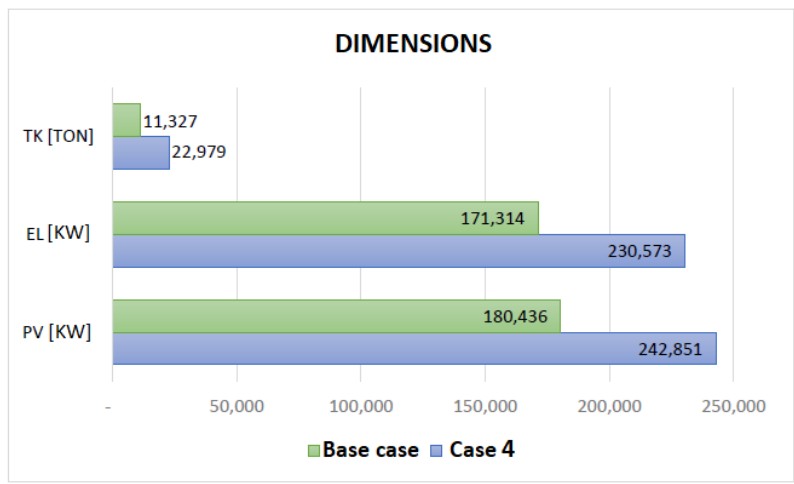

**Figure 9.** Comparison base case and case 4.

### 3.5. Case Study 5: Total CAPEX Variation

This case study consisted of applying all the reductions previously described; that is, decreases in CAPEX PV, CAPEX electrolyser, CAPEX storage, and PPA. The program showed that by reducing all CAPEX parameters and the cost of the electrical network, the photovoltaic plant used a capacity of 247.33 MW, the electrolysis of 234.82 MW, and storage tanks equivalent to 24,771 tons; results very similar to the previous cases. However, the total cost was significantly affected, giving a value of 356,406,598 USD. For this case, the amount of hydrogen produced did not change either; therefore, the NPV *H*2 was equivalent to 136,180,922 kg. In this way, it was demonstrated that the LCOH for this case study was 2.62 USD/kg. As in the other cases, in Figure 10 it can be seen that the capacities of the equipment were greater than those of the base case. However, the LCOH of this case study was considerably lower than the other cases studied.

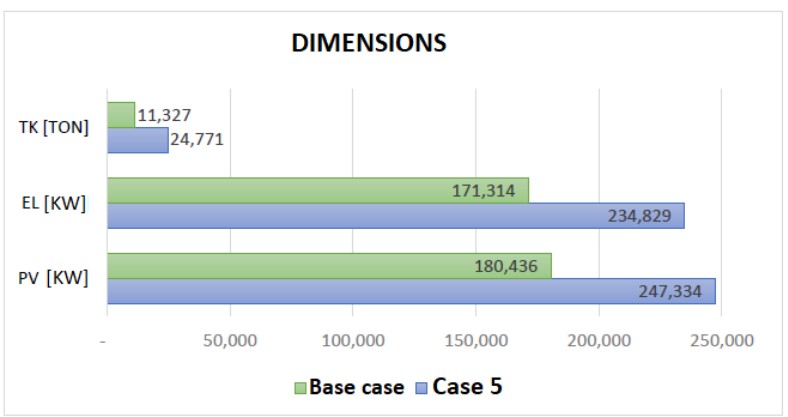

**Figure 10.** Comparison of the base case and case 5.

### 4. Discussion

All the cases analyzed considered a constant hydrogen flow demand of 1826 kg/h, this being a representation of the hydrogen flow in the port of Mejillones. Table 7 shows the values of the equipment capacities, according to the technology involved, for each of the cases studied.

**Table 7.** Comparison of the capabilities and LCOH for the case studies.

| Case Study | Solar Plant | Electrolyser | Storage | LCOH |
|---|---|---|---|---|
| Base case | 180 MW | 171 MW | 11 ton | 3.5 USD/kg |
| Case 1 | 246 MW | 234 MW | 24 tons | 3.25 USD/kg |
| Case 2 | 108 MW | 103 MW | 0 ton | 3.18 USD/kg |
| Case 3 | 247 MW | 235 MW | 24.7 tons | 2.98 USD/kg |
| Case 4 | 242 MW | 230 MW | 22 tons | 3.48 USD/kg |
| Case 5 | 247 MW | 235 MW | 24.7 tons | 2.62 USD/kg |

It can be observed that by decreasing the CAPEX of the photovoltaic system (case 1), it was possible to reduce the LCOH by 7.12%, since the use of energy from the electrical network was decreased, due to the increase in capacity of the photovoltaic system, electrolyser, and storage. Additionally, it can be seen that by reducing the cost of energy from the electrical grid (case 2), the LCOH was even lower than in case 1. On the other hand, by decreasing the CAPEX of the electrolyser (case 3), it can be observed that the LCOH decreased by approximately 15% with respect to the base case. However, when reducing the CAPEX of the storage system (case 4), this did not present a considerable variation with respect to the base case. Finally, when considering all the reductions in CAPEX and the energy cost of the network (case 5), the lowest LCOH of all the cases analyzed was observed, decreasing by 25% compared to the base case.

## 5. Conclusions

The optimization model presented in this research considers a photovoltaic plant located in the Atacama desert, approximately 200 km from the port of Mejillones, with a 35% on-grid plant factor; since, where there is no solar radiation, energy is consumed from the electrical network, through a green PPA (energy comes from renewable sources). The optimization program was carried out in AIMMS software, considering that 15% more photovoltaic energy is generated in summer than in winter. From this, a dimensioning of the photovoltaic system of 180 MW was obtained, an electrolyser capacity of 171 MW and a storage tank of 11 tons of hydrogen. These data represented the base case results.

On the other hand, case 1 consisted of reducing the CAPEX of the photovoltaic system by 16%, from which an LCOH of 3.25 USD/kg was obtained, as well as an increase of the photovoltaic system by 246 MW, the capacity of the electrolyser by 234 MW, and the size of the tank by 24 tons of gaseous hydrogen. However, the consumption of the electrical network decreased by 55% with respect to the base case.

Then, case 2 was analyzed, which consisted of reducing the energy contract price (PPA) with respect to the base case; with this, an LCOH of 3.18 USD/kg was obtained; that is, a lower value than the base case and case 1. This is because hydrogen is not stored, because it is more convenient to consume from the grid and from the photovoltaic system without investing in a storage system. In addition, it was shown that the photovoltaic plant had a smaller dimensioning of 108 MW; that is, 40% lower than the base case. In fact, the annual energy consumption of the network for case 2 was 576 GWh; that is, a 60% increase with respect to the base case (359 GWh), and the annual photovoltaic energy consumption decreased by 40% with respect to the base case.

For case 3, the CAPEX of the electrolyser was decreased and an LCOH of 2.98 USD/kg was obtained, corresponding to the lowest value of all the cases analyzed (except case 5). Regarding the sizing of the equipment, similar values to case 1 were obtained.

On the other hand, case 4 was analyzed, where the CAPEX of the storage tank was reduced. From this, an LCOH of 3.48 USD/kg was obtained, a value very close to the base case. Therefore, this input data did not greatly influence the minimization of the LCOH costs. The sizing of the equipment corresponded to a 242 MW photovoltaic plant, a 230 MW electrolyser, and a 22-ton hydrogen tank.

Finally, case 5 consisted of a decrease in the CAPEX of the systems and the price of the energy contract. From this, an LCOH of 2.62 USD/kg was obtained, which represents a projection towards 97 in the year 2030. It should be noted that the dimensioning of the photovoltaic system, the electrolyser, and the tank were the same as in case 3; that is to say, the most influential factor in minimizing the LCOH was the CAPEX of the electrolyser (case 3).

From this investigation, it can be deduced that the final cost of hydrogen production depended by approximately 60% on the value of the electricity generated that fed the electrolyser, 20% on the CAPEX of the electrolyser, 15% for the rest of the installation, and 5% on the concept of operation and maintenance. As previously shown, it was found that the element that mainly affects the production costs of green hydrogen is the cost of electricity generation. However, considering the study cases analyzed in this research work, it can be seen that the most influential factor in minimizing the LCOH consists of the CAPEX of the electrolyser, demonstrated by means of the sensitivity analysis in case 3. Regarding the results optimized using the AIMMS program, for all the cases analyzed, the LCOH was calculated at a constant hydrogen flow demand of 1826 kg/h, which represents the final hydrogen flow in the port of Mejillones. For the base case with the parameters of the year 2018–2019, an LCOH of 3.5 USD/kg was obtained; that is, an acceptable value within the range described by the International Energy Agency of 3–7.5 USD/kg for the year 2020 [16].

An important point corresponds to the consumption of water. In this context, the use of water to produce green hydrogen is projected at 0.6% of consumptive use of water at the national level; a figure that does not compromise the use of water resources, compared to other productive sectors [20]. On the other hand, the national consumption in Chile is of

the order of 4,900,000 L per second, and its consumptive use is 7%. In relation to the base case analyzed, 145,267,616 L per year would be needed; that is, on average 4.6 L per second, to produce 16 kton of hydrogen per year. Therefore, the water for electrolysis would not compromise the natural resources of the country.

Hydrogen has been widely studied, not only in Chile, but also in many countries around the world. It appears from this research that this is just the beginning for this global fuel.

The search to achieve the greatest hydrogen production, but at a low cost, is the current task that researchers are focusing on, both to generate an export business and for domestic consumption. Optimizing each of the set of elements that makes it possible to produce hydrogen has become an essential task for achieving this goal.

Based on this information, Chile is seen as an important country in the production of hydrogen, thanks to its development in terms of renewable resources, which are continuously growing. As one of the great barriers to hydrogen production is the cost of energy, countries that have a large amount of renewable sources have the potential to be producers of hydrogen. Faced with this, Chile has great opportunities to become one of the largest exporters of hydrogen, because it will be able to produce much more than needed for the country's internal consumption.

Currently, green hydrogen is not profitable, so this study on a model optimizer to produce photovoltaic hydrogen for subsequent export from the port of Mejillones in Chile will allow us to obtain a clearer vision of the capabilities of the production systems for hydrogen designed in Chile, in order to be considered as an export business.

**Author Contributions:** Conceptualization, methodology, investigation, writing, review and editing, M.T.M.D.; conceptualization, methodology, investigation, J.G.; review and editing, H.C.O. All authors have read and agreed to the published version of the manuscript.

**Funding:** This research was funded by FONDEF IT20I0126.

**Data Availability Statement:** Not applicable.

**Conflicts of Interest:** The authors declare no conflict of interest.

## Abbreviations

The following abbreviations are used in this manuscript:

| | |
|---|---|
| GEI | Greenhouse gases |
| ERNC | Non-conventional Renewable Energies |
| CAPEX | Capital Expenditure |
| OPEX | Operational Expenditures |
| LCOH | Levelized Cost of Green Hydrogen |
| VAN | Net Present Value |
| PPA | Power Purchase Agreement |

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
