# Peer review of "Economic Analysis: Green Hydrogen Production Systems"

_processes, doi:10.3390/pr11051390_

Round 1

Reviewer 1 Report

The paper mainly deals with the economical analysis of green hydrogen production system. Some issues should be addressed properly for the re-evaluation of the paper.

(1)   The knowledge gap should be supplemented.

(2)   The novelty of this paper should be stated exactly.

(3)   Page 6, correcting Table ??. There are many similar errors in the whole text.

(4)   The result should be analyzed and discussed more quantitatively.

Author Response

Good afternoon, thanks for the comments. I send the new version.

Reviewer 2 Report

In conclusion, I recommended that the paper should be accepted after mayor revision, including the following changes:

1. Comparing the existed results, what are your novelty results? Could you compare your results with the other investigations in theory parts?

2. The motivation of this manuscript isn't clear. The authors are suggested to further illustrate the motivation. The frame of the method should be explained a bit.

3. Although the drawbacks of the existing methods have been discussed, the challenges of this work are suggested to be further clarified. The analysis of the extracted results should be extended.

4. The format of this manuscript should be further improved. There exist some parts should be improved. Please carefully check and revise the whole manuscript.

5. Literature section must be extended with more papers.

6. The literature contribution of the manuscript should be with more detailed in the introduction section.

7. Results section is insufficient. It must be given with more analysis.

8. Conclusion section should be extended with more verbal and numerical results.

Author Response

(The authors gave the same response as above.)

Reviewer 3 Report

The paper is about the cost estimation of green hydrogen production. The manuscript is within the scope of Processes. Also, green hydrogen production is an extremely hot topic nowadays because green hydrogen can be used as an energy carrier for the decarbonization of several sectors. However, the manuscript should be appropriately revised for a possible publication. My main concern related to the paper is about novelty of the paper. Many papers have been published to estimate cost of green hydrogen production. The authors should clearly explain novelty and obejctives of their work in the introduction. The authors should also cite related works in their manuscript. There are only 19 references cited in their manuscript. However, many papers are available in the literature about this topic. The abstract does not include any numerical values to show their key findings. The authors should also rewrite the abstract. The conclusion of the manuscript should also be improved.

Author Response

(The authors gave the same response as above.)

Round 2

Reviewer 1 Report

The paper has been revised carefully in terms of the review comment. It is recommended acceptance for the manuscript.

Reviewer 2 Report

The authors should explain better where is the innovativeness with respect to the state of the art in the literature. The literature review in the Introduction section deals with the topic, but is more general with respect to the particular piece of work proposed in this manuscript. Therefore, from the discussion in the Introduction I can’t really understand the added value of this work.

Authors should show a comparative analysis with other methods, and the results could be compared. The results should be more discussed. 

Figures must be improved.

The conclusions should indicate the data obtained in the results, and the comparison with other methods.

Reviewer 3 Report

The present paper can be beneficial for the research community. However, this version of the paper should also be revised by the authors for a potential publication. 

My specific comments are below:

  1. The authors can use energy carrier for hydrogen rather than fuel in the abstract and throughout the paper.
  2. The authors should revise the abstract. The abstract should include first motivation (why this is study important), a few sentences about the current literature, a few sentences about originality of their work. Also, the authors should briefly explain their methodology used in the present work. The last part of the abstract should include key numerical finding related to this work.
  3. The authors should expand the literature review in the introduction. There are many relevant papers about green hydrogen production. They can cite more papers in the introduction.
  4. The authors should clearly highlight their novelty and objectives in the intoruction. It should be explained how this paper contriubutes the current literature. 
  5. The authors can prepare a supplementary file or an Appendix to show clearly all input parameters used for modelling of the system. 
  6. The authors can compare their results with the similar studies in the literature. This can increase impact of the paper.
  7. The authors can add a few dimensionless indicators to show different performance indicators of the system such as grid indepence, utilization factor of solar energy etc.
